# Mapping the pathways of photo-induced ion migration in organic-inorganic hybrid halide perovskites

Taeyong Kim [1,2], Soyeon Park[3], Vasudevan Iyer[4], Basamat Shaheen [1], Usama Choudhry[1], Qi Jiang[3], Gage Eichman [4], Ryan Gnabasik[1], Kyle Kelley [4], Benjamin Lawrie [4,5] ✉, Kai Zhu [3] ✉ & Bolin Liao [1] ✉

Organic-inorganic hybrid perovskites exhibiting exceptional photovoltaic and optoelectronic properties are of fundamental and practical interest, owing to their tunability and low manufacturing cost. For practical applications, however, challenges such as material instability and the photocurrent hysteresis occurring in perovskite solar cells under light exposure need to be understood and addressed. While extensive investigations have suggested that ion migration is a plausible origin of these detrimental effects, detailed understanding of the ion migration pathways remains elusive. Here, we report the characterization of photo-induced ion migration in perovskites using in situ laser illumination inside a scanning electron microscope, coupled with secondary electron imaging, energy-dispersive X-ray spectroscopy and cathodoluminescence with varying primary electron energies. Using methylammonium lead iodide and formamidinium lead iodide as model systems, we observed photo-induced long-range migration of halide ions over hundreds of micrometers and elucidated the transport pathways of various ions both near the surface and inside the bulk of the samples, including a surprising finding of the vertical migration of lead ions. Our study provides insights into ion migration processes in perovskites that can aid perovskite material design and processing in future applications.

Rapid progress has been made in the past decade in improving the power conversion efficiency of photovoltaic cells based on organic-inorganic hybrid perovskites (OIHPs). Recent reports have described power conversion efficiencies exceeding 25%[1–4]. The remarkable performance has been attributed to OIHPs' unique optoelectronic properties such as a direct band gap and sharp absorption edge near visible wavelengths[5–7], large photocarrier diffusion lengths[8,9] and high tolerance to defects and grain boundaries[10,11], which are key factors for photovoltaic cells[12–14]. Additionally, OIHPs have shown other interesting properties, such as their tunability[15], moderate mobilities of charge carriers[8] and ions[16], strong polaronic coupling[13,17], and a prominent photostriction effect[18]. Coupled with their cost-effective low-temperature solution processability[19], these unusual optoelectronic properties suggest wide-ranging practical optoelectronic applications of OIHPs beyond solar cells.

In addition to investigations of device efficiency optimization, extensive prior studies have focused on understanding and improving the stability of OIHPs[20], which is a crucial limiting factor for long-term

[1]Department of Mechanical Engineering, University of California, Santa Barbara, CA 93106, USA. [2]Department of Mechanical Engineering, Seoul National University, Seoul 08826, Republic of Korea. [3]National Renewable Energy Laboratory, Golden, CO 80401, USA. [4]Center for Nanophase Materials Sciences, Oak Ridge National Laboratory, Oak Ridge, TN 37830, USA. [5]Materials Science and Technology Division, Oak Ridge National Laboratory, Oak Ridge, TN 37830, USA. ✉e-mail: lawriebj@ornl.gov; kai.zhu@nrel.gov; bliao@ucsb.edu

practical applications. It has been well established experimentally that many extrinsic factors can lead to degradation in these hybrid perovskite materials[20,21], including exposure to moisture[21,22] and oxygen[23], thermal decomposition[24], and light-induced chemical reactions[25,26]. Corresponding strategies have been developed to counter some of the degradation factors, e.g. OIHP encapsulation to avoid exposure to moisture or oxygen[27,28].

Many of the instability factors are related to the weak chemical bonds[29] and strongly anharmonic lattice dynamics[30] in OIHPs that also lead to ultralow thermal conductivity[31] and facile ion migration induced by temperature[32], electric fields[33], or light[34]. Since temperature gradients, electric fields, and light exposure are unavoidable in the normal operation of solar cells, it is recognized that ion migration processes play an essential role in OIHP-based devices[35,36]. In this light, recent studies have observed ion-migration occurring in OIHPs and its impact on the material morphology[33], composition and device stability[26,35,37]. In addition, these studies have reported that ion migration might cause the abnormal photocurrent hysteresis effect, which, in turn, affects fill factors and solar cell device performance[37]. Unlike extrinsic factors such as moisture and oxygen, ion migration is intrinsic in OIHPs, likely linked to certain strongly anharmonic lattice vibration modes involving the mobile ions[38], and, therefore, requires more sophisticated mitigation strategies[36,39]. On the other hand, ion migration represents a possible mechanism for the large static dielectric constant in OIHPs[40] that can effectively screen charged defects and benefit charge transport. Some studies have also suggested that the redistribution of ions as a result of migration can lead to optimized p-i-n structures[41] or healed charge trapping centers[42] and improve the device performance. Besides, efficient ion migration in OIHPs also implies potential applications of these materials as ionic conductors[34]. These examples highlight the importance of understanding the ion migration processes in OIHPs[35].

Previous works have reported light-induced migration of ions such as the halide anion ($I^-$), the metal cation ($Pb^{2+}$), and the organic cations (MA$^+$ = Methylammonium, FA$^+$ = Formamidinium)[34,43–46]. In parallel, prior computational studies have shown that the migration rate is determined by the activation energy and that $I^-$ is more likely to migrate owing to its lowest activation energy among other constituent ions ($I^-$: 0.08−0.58 eV in refs. [47–50]; MA$^+$: 0.46−0.84 eV in refs. [47–50]; $Pb^{2+}$: 0.8−2.31 eV in refs. [47,48]). Experimental measurements have corroborated this, with the diffusivity of $I^-$ (~$10^{-12}$ to ~$10^{-9}$ cm$^2$ s$^{-1}$[47,50,51]) far exceeding that of organic cations (MA: ~$10^{-16}$ to ~$10^{-11}$ cm$^2$ s$^{-1}$[47,50,51]). In most previous experimental studies, the $Pb^{2+}$ ion was assumed to be immobile up to available detection limits[52]. Despite the abundant previous literature on photo-induced ion migration in OIHPs, the detailed microscopic migration pathways of various ions remain unclear. In particular, ion migration both parallel to the sample surface (lateral ion migration) and perpendicular to the sample surface (vertical ion migration) can occur given the light intensity variation along the lateral direction and the finite optical absorption depth into the sample. Therefore, experimental tools that are sensitive to the distribution of ions along both the lateral and vertical directions are highly desirable.

In this work, we focus on photo-induced ion migration in OIHPs. We incorporated a laser source in situ into a scanning electron microscope (SEM) and combined secondary electron imaging (SEI), energy-dispersive X-ray spectroscopy (EDS) and cathodoluminescence (CL) with different primary electron (PE) energies to directly map the three-dimensional (3D) ion migration pathways in two archetypal OIHPs: MAPbI$_3$ and FAPbI$_3$. We demonstrated that both lateral and vertical migration of various ion species driven by light exposure can be mapped in this manner, which further correlates to the optoelectronic properties of OIHPs. This information is critical for designing more stable and efficient devices based on OIHPs.

## Results

### Probing ion distribution within different depths

Our experimental setup for the SEI and EDS measurements is hosted at the University of California Santa Barbara (UCSB); it is identical to that described in ref. [53], and the principle of operation is schematically illustrated in Fig. 1A. An optical beam from a fiber laser (Clark-MXR IMPULSE, photon energy: ~2.4 eV; repetition rate: 5 MHz) was coupled into the SEM sample chamber and focused onto the sample, with a beam diameter of 50 μm. To examine the impact of the photo-induced ion migration, the optical power and the exposure time were varied, ranging from ~2.5 to ~6.7 mW (optical fluence: ~39.8–106.6 μJ cm$^{-2}$), and from 5 to 180 min, respectively. These optical parameters for the SEI and EDS measurements were chosen to minimize the laser-induced heating effect, and we estimated that the highest laser-induced temperature rise in our experiment was within 30 K (see Supplementary Information Note 5). Then, changes in the secondary electron (SE) emission and the elemental composition induced by the optical exposure were mapped in situ with concurrent SEI and EDS mapping. The depth-dependent ion distribution was probed by comparing the response to PEs with different kinetic energies (30 keV and 5 keV) and different penetration depths into the sample. The CL measurements coupled with in situ light-exposure were conducted at the Center for Nanophase Materials Sciences (CNMS) at the Oak Ridge National Laboratory (ORNL), and the corresponding experimental details are given in Methods. The procedure for sample fabrication is also described in Methods. The structure of the sample stack is illustrated in the inset of Fig. 1D. More characterization data (X-ray photoelectron spectroscopy, X-ray diffraction and photoluminescence) to confirm the sample quality is provided in the Supplementary Information Note 1A. A high vacuum of at least $1 \times 10^{-6}$ torr was maintained inside the SEM chambers for all measurements reported here, thus minimizing the influence of moisture or oxygen. Representative EDS spectra of a pristine FAPbI$_3$ sample deposited on a fluorine-tin-oxide (FTO) coated glass are given in Fig. 1B (the MAPbI$_3$ spectra are given in the Supplementary Information Note 1B). In the EDS spectra, peaks corresponding to characteristic X-ray lines of various chemical elements can be seen. The intensity of these characteristic peaks also depends on the PE kinetic energy used. With 30-keV PEs, prominent peaks near 1.74 keV, 2.35 keV, 3.44 keV, and 3.94 keV were attributed to X-ray emission from silicon ($K_\alpha$ band), lead ($M_\alpha$ band), tin ($L_\alpha$ band) and iodine ($L_{\alpha 1}$ band)[54–57]. The lead and iodine signals originated from the FAPbI$_3$ layer, while the tin and silicon only existed in the substrate, signaling a deep penetration of the 30-keV PEs. In contrast, with 5-keV PEs, we found that the tin and iodine bands were suppressed, the lead band remained, while other lower-energy peaks emerged near 277 eV, 392 eV, and 525 eV, corresponding to the characteristic X-ray emission from carbon, nitrogen and oxygen[58]. Additional peaks associated with tin and iodine in the spectra can be observed, e.g. the tin $L_{\beta_1}$ line (3.66 keV), while the peak at ~4.2 keV can be attributed to the tin $L_{\gamma_1}$ line, 4.13 keV) and the iodine $L_{\beta_1}$ line (4.22 keV), as labeled in Fig. 1B. These lines were not used in the following analysis due to redundancy. Additional EDS spectra with intermediate PE energies (10 keV and 20 keV) are shown in the Supplementary Information Note 1C.

To quantitatively understand the probing depth of PEs with different kinetic energies, we conducted Monte Carlo simulations of the PE trajectories with 30-keV and 5-keV kinetic energies; the results are shown in Fig. 1C, D. The detailed parameters used in our simulation are described in the Supplementary Information Note 2. As shown in Fig. 1C, D, many of the 30-keV PEs penetrated the entire FAPbI$_3$ layer and into the substrate, while those at 5 keV were mostly confined within 200 nm near the surface of FAPbI$_3$. This indicates that the EDS spectra obtained with PEs with different kinetic energies are sensitive to ion distributions at different depths in the sample. Additionally, we note here that the self-absorption depth of the X-ray photons emitted inside the samples was comparable to or greater than the sample

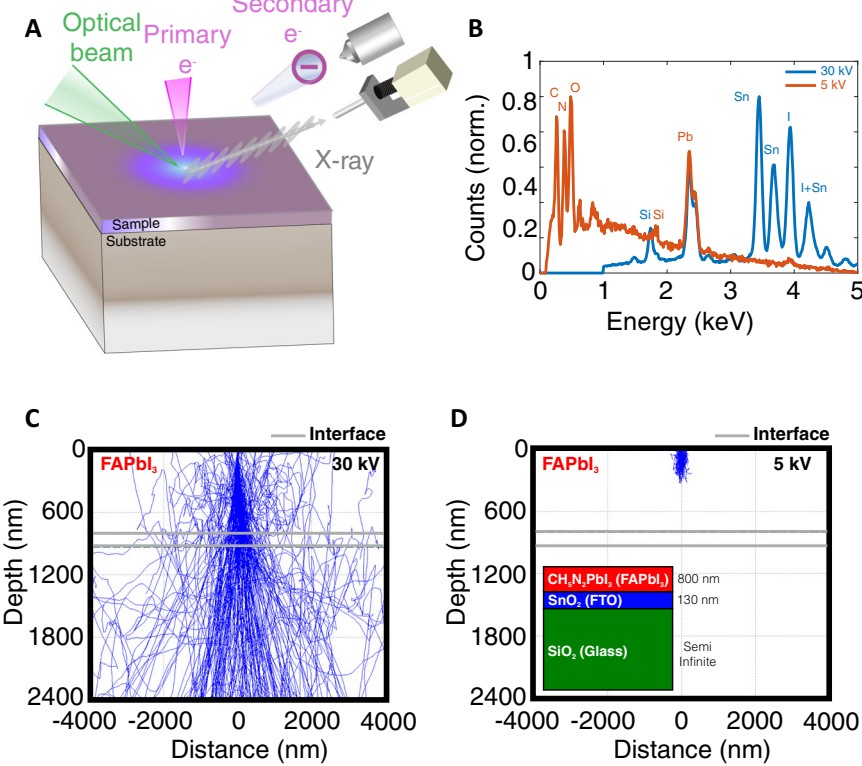

**Fig. 1 | Schematic illustration of the experimental setup and sample structure.** **A** Illustration of the SEM experimental platform incorporating an in situ laser beam with SEI and EDS detectors. **B** The measured EDS spectra of FAPbI₃ without light exposure at the primary electron energies of 30 keV (blue line) and 5 keV (red line). The corresponding elements are annotated near each characteristic peak. Also shown are the simulated trajectories of primary electrons with energies of (**C**)

30 keV, and (**D**) 5 keV. While the primary electrons can penetrate the entire sample stack at 30 keV, those at 5 keV are mostly confined within 200 nm near the FAPbI₃ surface. The interfaces between FAPbI₃ and fluorine doped tin oxide (FTO), and between FTO and the glass substrate are labeled with gray lines. The inset in **D** shows the structure of the FAPbI₃ sample stack.

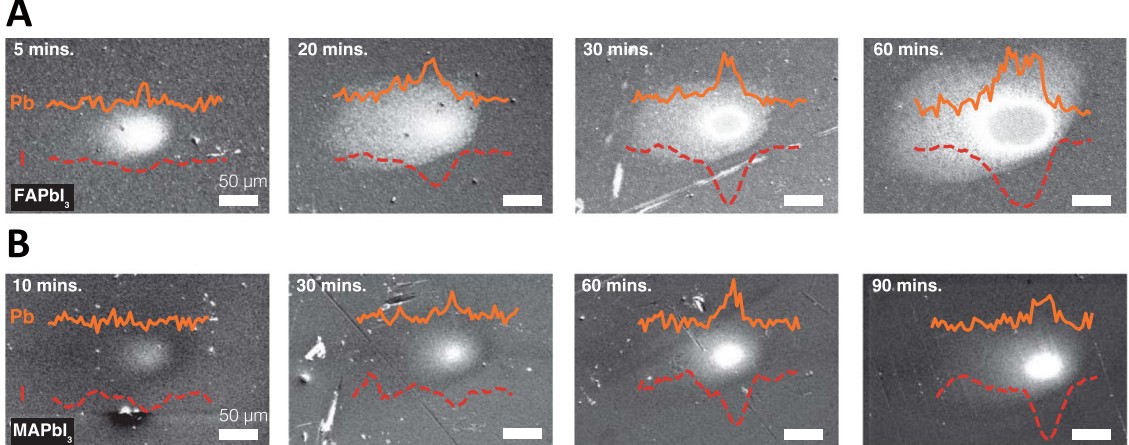

**Fig. 2 | Secondary electron images with line profiles of iodine and lead distribution after optical beam exposure.** **A** SE images of FAPbI₃ after laser illumination for 5–60 min. **B** SE images of MAPbI₃ after laser illumination for 10–90 min. The red dashed lines (orange solid lines) indicate the X-ray counts for iodine (lead) at 3.94 keV (2.35 keV) across the illuminated area as measured using EDS. The

iodine line was probed by 30-keV PEs and the lead line was probed by 5-keV PEs. For all the data presented here, an optical fluence of ~106.6 µJ cm⁻² was used. To minimize the potential sample damage due to the electron beam exposure, a new location on the sample was chosen for each of the light-exposure experiments.

thickness, except for those X-ray photons emitted from carbon, nitrogen and oxygen (see the Supplementary Information Note 3 for further details), indicating that our measurement was sensitive to the PE penetration depth rather than the X-ray absorption depth except for carbon, nitrogen and oxygen.

Our experimental configuration permitted us to characterize the photo-induced ion movement along with their impact on the SE yield

in an in situ manner. Figure 2 shows several SE images of FAPbI₃ and MAPbI₃ samples, which were taken after the laser beam exposure (optical fluence: 106.6 µJ cm⁻² for both FAPbI₃ and MAPbI₃) at the center of the field of view for variable laser dwell times. To minimize the potential sample damage due to the electron beam exposure, a new location on the sample was chosen for each of the light-exposure experiments reported here. As shown in Fig. 2A, for FAPbI₃, the SE

image contrast became brighter after the laser exposure for 5 minutes. Further exposure of up to 20 min led to the bright contrast expanding beyond the laser spot size. Under prolonged optical exposure of ≥30 min, an area with dark contrast emerged near the center of the beam area, forming a ring-shaped profile that was surrounded by bright contrast at the outer region. We also observed that additional optical exposure caused further expansion of the bright ring, while the surrounding bright contrast area far exceeded the size of the optical beam. In contrast, the SE images of MAPbI$_3$ after light exposure exhibited a slower temporal evolution of the bright contrast. The size of the area with bright contrast after longer exposure times was observed to be comparable to the optical beam size. This observation is consistent with a previous study of photo-induced secondary elec-tron contrast in MAPbI$_3$[59]. At lower optical fluences, qualitatively similar contrast in SE images was seen, although the required exposure time was much longer (see the Supplementary Information Note 1D for additional data). Further tests suggest that the photo-induced sec-ondary-electron contrast remains after a long time, suggesting the photo-induced effect is irreversible, likely due to iodine loss to the vacuum (see the Supplementary Information Note 1E for additional data). High-resolution SEM images show that the light-exposed area has smaller grains compared to the pristine area (see Supplementary Information Note 1D for additional data). We can further correlate the observed bright SE contrast with the ion distribution measured in situ by EDS. In particular, the X-ray counts corresponding to the char-acteristic iodine line at 3.94 keV (probed by 30-keV PEs) and lead line at 2.35 keV (probed by 5-keV PEs) were measured along a horizontal line across the center of the area exposed to the optical beam and were given as red dashed lines in Fig. 2A, B. In general, we observed a decrease of iodine counts within the area exposed to light, and the area with an iodine deficiency increased with prolonged light exposure. Particularly in FAPbI$_3$ (Fig. 2A), the iodine concentration profile agreed well with the spatial extent of the bright contrast, suggesting long-ranged migration of iodine ions far outside the illuminated region. However, the iodine concentration profile within the area illuminated by light showed a Gaussian shape, which does not explain the bright ring-shaped contrast observed in FAPbI$_3$. In contrast, although the lead concentration profile did not correlate to the overall spatial range of the bright contrast, a two-peak structure developed in FAPbI$_3$ after 30-min light exposure that matches the bright ring-shaped contrast, indicating a lead-rich phase was responsible for the significantly enhanced SE emission from the ring-shaped area. With our current technique, however, the precise composition of this lead-rich phase could not be determined, since the 30-keV PEs probed the iodine concentration inside the entire thickness of the sample, not only at the surface. We note that deQuilettes et al. observed a ring-shaped redis-tribution of iodine ions near the surface of MAPbI$_3$ using time-of-flight secondary-ion-mass spectroscopy after light soaking of the sample[46], while no migration of lead ions was detected.

**Quantification of ion migration processes**

We can quantify the ion migration processes by carefully examining the ion distributions mapped by EDS after different light exposure times. We first focus on the iodine ions, which are known to be the most mobile species in OIHPs. In Fig. 3A, we show the changes in the characteristic iodine X-ray photons at 3.94 keV from FAPbI$_3$ near the illuminated region. The changes in the counts were calculated relative to the value measured in the pristine region without light exposure and then normalized by the counts from the pristine region. The changes in counts were measured using EDS with 30-keV PEs and plotted as a function of the distance from the center of the optical beam and after different light exposure durations. A reduction of the iodine X-ray counts was observed in the illumi-nated region, signaling a deficiency of iodine as a result of light exposure. Furthermore, the iodine deficiency kept increasing and

its spatial distribution kept expanding as a function of exposure time. Since the 30-keV PEs probed the entire sample thickness, the observed iodine deficiency indicated the amount of iodine inside the sample was reduced, likely due to the formation of I$_2$ vapor escaping to the vacuum[60]. In addition, the expanding spatial profile of the iodine deficiency signals migration of iodine ions over a lengthscale of a few hundred micrometers, significantly exceeding the optical beam size, driven by the iodine concentration gradient due to the I$_2$ escaping from the surface. From the time dependence of the spatial profile of the iodine deficiency, we can estimate the diffusivity of iodine ions, as shown in Fig. 3B. The extracted iodine ion diffusivity is around $3.5 \times 10^{-10}$ cm$^2$ s$^{-1}$, which is within the range of previously reported values[47,50,51]. A detailed analysis of the diffu-sion process is described in the Supplementary Information Note 4. We note that the value of the halide diffusivity measured in this work is two to three orders of magnitude higher than that measured in single crystalline perovskites even at an elevated temperature (100° C)[32]. The most likely cause of this discrepancy is the presence of grain boundaries, which were previously shown to significantly facilitate the migration of ions in polycrystalline perovskite thin films[61].

Next, we focus on the lead ions, whose distribution can be map-ped by PEs with both 30-keV and 5-keV kinetic energies. As shown in Fig. 3C, D, the lead ion distribution probed by 30-keV PEs remained almost unchanged as a result of light exposure, while its distribution probed by 5-keV PEs showed a significant increase with light exposure. Since the 30-keV PEs can probe the entire sample thickness and the 5-keV PEs only probe near the sample surface, this surprising finding implied that the migration of lead ions mainly occurred along the vertical direction, namely moving from inside the sample towards the sample surface. More interestingly, the lead distribution near the surface as probed by the 5-keV PEs showed a double-peak structure after a long exposure, consistent with the bright ring contrast observed in the SE images shown in Fig. 2A. As discussed before, this suggests that a lead-rich phase is responsible for the enhanced SE emission in the bright ring region.

Another surprising finding was the observed increase of the X-ray counts associated with silicon and tin, as shown in Fig. 3E, F. While a small trace of silicon was detected with the 5-keV PEs (as shown in Fig. 1B), the silicon distribution did not change appreciably due to the light exposure (See the Supplementary Information Note 1F for addi-tional data). However, the silicon and tin distributions as probed by the 30-keV PEs showed a significant increase with prolonged light expo-sure, suggesting increased silicon and tin concentration within the region probed by 30-keV PEs. One possibility is that the density of the perovskite film was changed, or structural changes created gaps in the perovskite film after light exposure that allowed more PEs to reach the substrate, increasing the X-ray emission from silicon and tin within the substrate. To investigate this possibility, we conducted a comparative study of FAPbI$_3$ deposited on a gold-coated FTO glass substrate (see Supplementary Information Note 1G for additional data and discus-sion). We did not observe any appreciable increase of X-ray counts associated with gold and tin after light exposure, so it is unlikely that more PEs reaching the substrate was the cause for the increased silicon and tin signal. An alternative explanation is the migration of silicon and tin ions induced by the isovalent lead vacancies created deep inside the bulk of the FAPbI$_3$ sample after the lead ions migrated towards the sample surface. This observation implies that ion migration can lead to unwanted interactions between the OIHPs and the charge transport layers that can be detrimental to solar cells based on OIHPs[62]. However, we do not have direct evidence to support the presence of silicon and tin ions within the perovskite layer after light exposure at this stage, which will be explored in a future work. Lastly, we also observed a decrease of nitrogen, carbon and oxygen concentration near the sample surface as probed by 5-keV PEs due to light exposure (See the

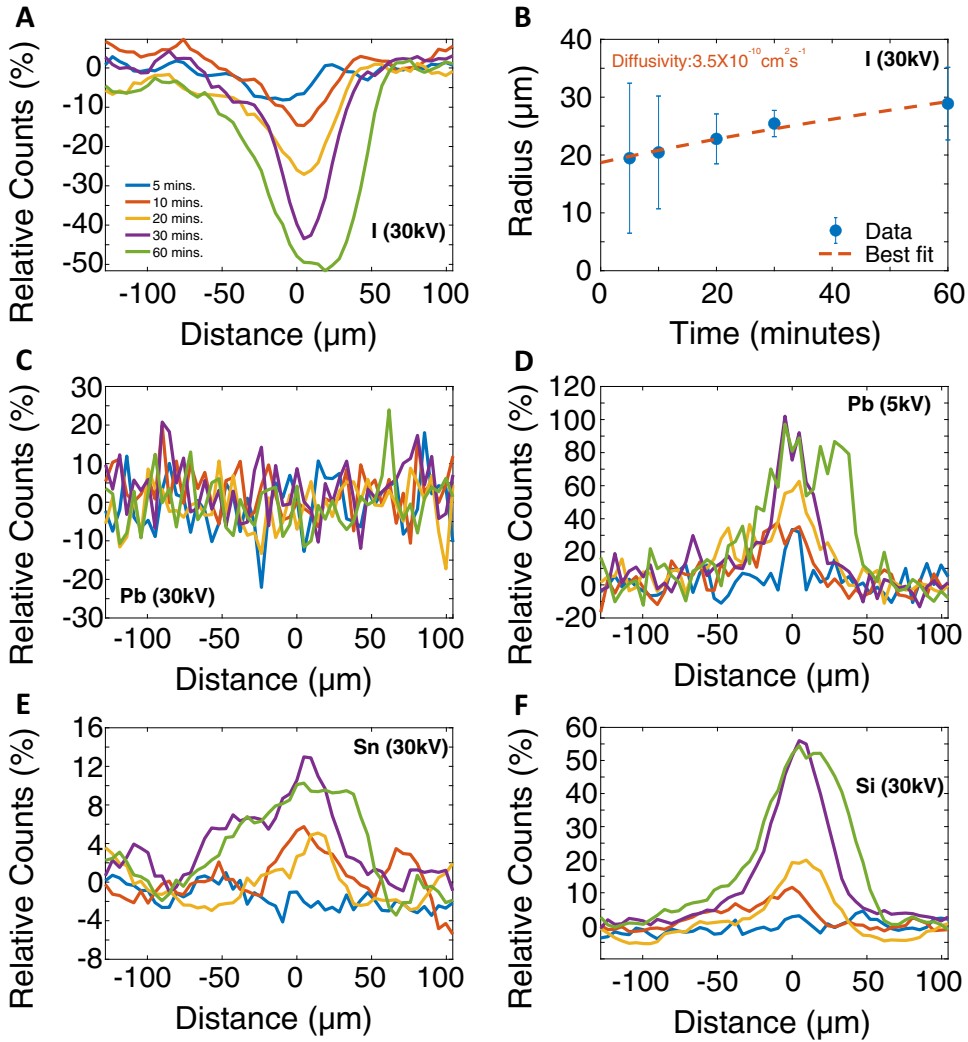

**Fig. 3 | EDS elemental analysis of FAPbI₃ after optical exposure.** An optical fluence of ~106.6 μJ cm⁻² was used for this dataset. **A** The change in the X-ray counts corresponding to iodine as probed by 30-keV PEs. The change in counts was calculated relative to the value measured from pristine areas without light exposure. **B** The corresponding fitted radii of the iodine deficiency distributions versus the exposure times measured by 30-keV PEs. The error bar indicates numerically determined 95% confidence intervals. Also shown are the measured EDS linescan datasets for (**C**) lead measured by 30-keV PEs, (**D**) lead measured by 5-keV PEs, (**E**) tin measured by 30-keV PEs, and (**F**) silicon measured by 30-keV PEs. All changes in X-ray counts are normalized by the pristine value before light exposure.

Supplementary Information Note 1F for additional data). However, we could not quantify the diffusivity of the organic ions due to the relatively low EDS sensitivity and, thus, a low signal-to-noise ratio.

Based on the observations described above, we summarize the microscopic picture of photo-induced ion migration in FAPbI₃ as follows. Photo-induced iodine deficiency near the surface creates an iodine concentration gradient that drives the diffusion of iodine ions towards the surface. The lateral iodine concentration gradient created by the Gaussian distribution of the laser beam intensity further induces iodine ion migration towards the center of the laser illuminated area. In the meantime, lead ions also migrate from the bulk towards the surface. Since lead cannot escape the sample surface, the accumulation of lead near the center of the illuminated area starts to drive lead ions to move outwards near the surface after longer light exposure, creating the double-peak lead concentration profile that corresponds to the bright ring contrast in SE images. As a consequence of the lead ion migration, there is a possibility that the lead vacancies left behind deep inside the sample bulk might induce the migration of isovalent tin and silicon ions inside the substrate and towards the sample, leading to an enhanced silicon and tin X-ray count, although we currently do not have direct evidence of their presence inside the perovskite sample.

## Correlating ion migration with cathodoluminescence
We note here that Barbé et al. observed a light-induced ring-shaped area with enhanced photoluminescence[45] in MAPbI₃ and they attributed it to the formation of a thin layer (<20 nm) of PbI₂ using Raman spectroscopy. However, no direct evidence in the form of photoluminescence near the bandgap of PbI₂ (~500–540 nm) was observed (due to a 532-nm excitation laser and weak PbI₂ PL at ambient temperatures[63]). CL microscopy can be used to effectively probe spatial heterogeneities in perovskite excitonic and defect emission, so we used CL microscopy with in situ laser exposure to better understand the effects of photo-induced ion migration on the optoelectronic properties of perovskite thin films.

For this measurement, a femtosecond laser with 495 nm wavelength and variable power was focused to a 5-μm spot on each film using a parabolic mirror in the SEM. A similar experimental design was previously used to study the photo-induced phase separation of hybrid halide perovskites[64,65]. The optical power used here (100 μW) was chosen to ensure the laser-induced temperature rise was below 30 K. The same parabolic mirror was used to collect CL generated by the perovskite films immediately after laser exposure. Notably, substantial research efforts in recent years have probed ion migration and

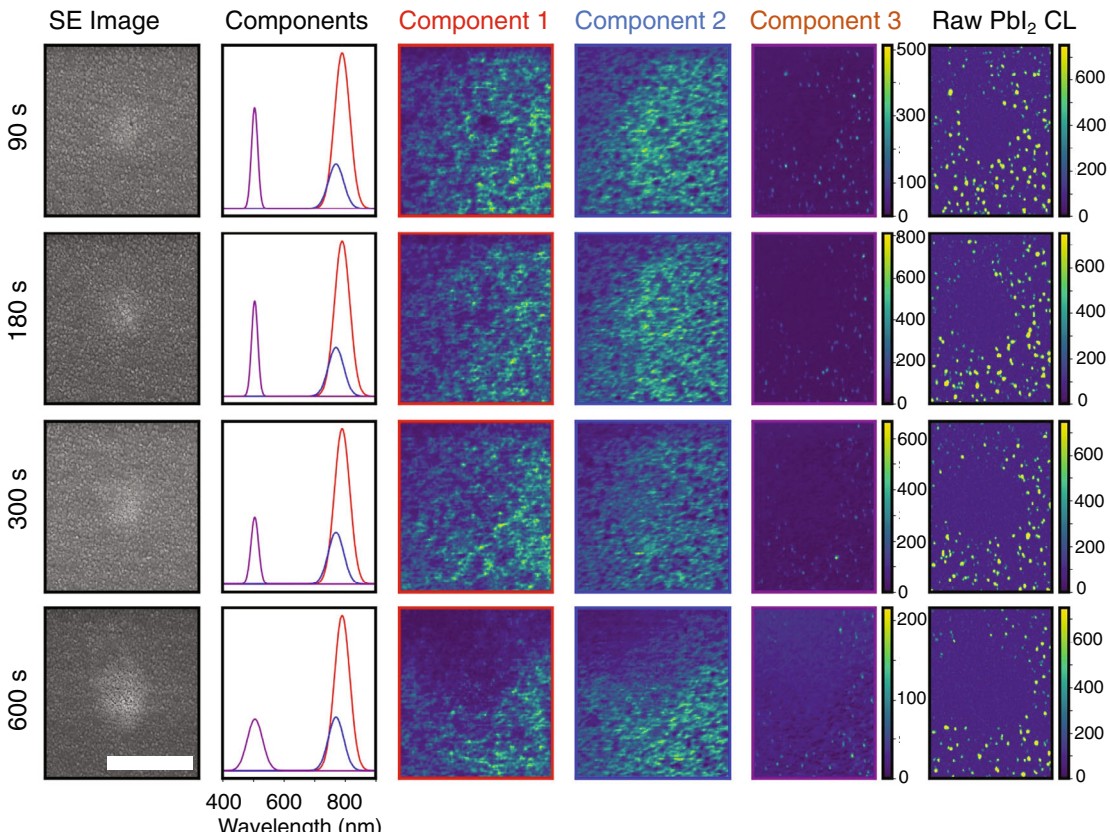

**Fig. 4 | Cathodoluminescence analysis of MAPbI₃ film after in situ laser exposure.** Correlated SE images (first column) and CL NMF decomposition (columns 2–5) after exposure to a 495-nm, 100-µW fs laser source focused to a 5-µm spot size for laser exposure times of 90–600 s. Three spectral components are used in the NMF decomposition to highlight the MAPbI₃ exciton CL (component 1; red), the partially degraded MAPbI₃ exciton CL (component 2; blue), and the PbI₂ CL (component 3; purple). While the NMF decomposition successfully distinguishes components 1 and 2, the PbI₂ CL is orders of magnitude weaker in intensity. The raw PbI₂ CL (integrated from 500-520 nm) is included in the final column for comparison. Scale bar: 5 µm.

perovskite decomposition under optical and electron-beam excitation and in the presence of various environmental factors[44,66–68]. Under the electron-beam conditions used here (5-keV PE and 7 pA of beam current), we have not observed substantial electron-beam induced degradation in past experiments[69].

The correlated SE and CL images acquired for different positions on a MAPbI₃ film as a function of laser exposure time are shown in Fig. 4. We used non-negative matrix factorization (NMF) to decompose the hyperspectral CL images into representative spectral components. The calculated explained variance ratio (EVR) is greater than 92% for all of the images acquired after laser exposure. The three spectral components for each image are shown in the second column of Fig. 4, while the spatial distribution of CL corresponding to the three spectral components are presented in columns 3–5. The raw PbI₂ CL (integrated from 500 to 520 nm) is included in the final column for comparison.

A few critical features are visible in Fig. 4. First, the SE images exhibit increased secondary electron brightness and changes in morphology after laser exposure that are consistent with those reported in Fig. 2 and Supplementary Fig. S7, and the size of the microstructured area increases with increasing laser exposure time. Second, CL NMF decomposition component 1 is consistent with excitonic luminescence, and component 2 is consistent with near-band-edge CL from an intermediate phase that results from partial degradation of the film[68,69]. Weak PbI₂ CL is visible in isolated spots across component 3, but the PbI₂ CL is orders of magnitude weaker than the excitonic perovskite CL. While the NMF decomposition successfully distinguishes between components 1 and 2, allowing for a clear understanding of the changes in excitonic states, it does not represent the PbI₂ CL as cleanly because of the substantial difference in signal intensity. The raw PbI₂ CL illustrated in the final column highlights an effect that is also visible in NMF component 3: the PbI₂ CL is completely suppressed in the area around the laser spot, and the area with suppressed PbI₂ CL increases monotonically with increasing laser exposure time. In contrast, while small changes to components 1 and 2 are observed near the laser spot, the laser exposure does not substantially detrimentally effect the perovskite excitonic luminescence. Thus, we can conclude that (a) the ion migration induced by laser exposure does not introduce any near-surface PbI₂ or partially decomposed OIHP phases with any statistical significance and (b) the ion migration completely suppresses PbI₂ defects in the vicinity of the laser spot.

At higher laser powers, thermal effects start to emerge, and the excitonic CL is also suppressed. Supplementary Figs. S16–S20 in the Supplementary Information highlight these thermal effects for MAPbI₃, FAPbI₃ and hybrid 97% FAPbI₃/3% MAPbBr₃ perovskite films. Critically, when the laser power was increased, the hybrid films exhibited substantially enhanced luminescence intensity in the area surrounding the laser spot. We observed similar results using a higher power continuous-wave laser source at 532 nm. Interestingly (as shown in the Supplementary Information Note 1I and Supplementary Fig. S21), we also observed a partial recovery of the excitonic CL emission near 800 nm in an environmentally degraded FAPbI₃ sample after light exposure, suggesting a beneficial impact of light-induced ion migration in degraded OIHP samples.

Lastly, we note here that thermally induced decomposition of OIHP thin films has been extensively reported previously[45,70,71]. Our observations in this work can be distinguished from thermal

decomposition for the following reasons: First, we observed long-ranged change of SE contrast (Fig. 2) far outside the beam-illuminated region. If the observed contrast change was due to thermal decomposition, it would be limited to the beam area (diameter 50 μm). The long-ranged change of SE contrast can only be explained by migration of iodine ions outside the illuminated area. Second, we observed a change of the absolute X-ray counts of lead near the surface of the OIHP samples. Thermal decomposition would not change the absolute lead content and, thus, was not expected to change the absolute X-ray counts of lead, which can, instead, be attributed to vertical migration of lead ions. Third and most importantly, we did not observe any spectral evidence of the thermal decomposition product $PbI_2$ in the CL measurement. It is well established that $PbI_2$ is the main solid-state product of thermal decomposition of both $MAPbI_3$ and $FAPbI_3$[70,71]. Indeed, from an intentionally degraded $FAPbI_3$ sample, we observed strong $PbI_2$ emission (Supplementary Fig. S21) while there was no $PbI_2$ signature in the light-exposed samples. This is a decisive evidence that the thermal decomposition in our experiments was minimal. Furthermore, thermal decomposition reported in previous works[70,71] typically occurred at temperatures higher than 100° C, while, in our experiments, great care was taken to limit the laser-induced temperature rise to be lower than 30 K. In addition, in our opinion, ion migration and decomposition are intrinsically intertwined processes that cannot be distinguished in an absolute sense. Any decomposition process will lead to a nonuniform distribution of ions that will drive ion migration. In the meantime, any ion migration process can also lead to local changes of the material composition that can be termed decomposition if the change is significant. In our case, due to the lack of spectral signature of the decomposition product $PbI_2$ and the small change of the CL components 1 and 2 in the exposed area (Fig. 4), we believe the composition change in our samples due to ion migration was small.

In summary, we utilized an in situ laser source incorporated into an SEM to directly map lateral and vertical photo-induced ion migration pathways in OIHPs with SEI, EDS and CL. We observed long-ranged iodine migration far beyond the illuminated region, a surprising vertical migration of lead ions and possible cross-boundary migration of silicon and tin ions from the substrate. CL analysis indicated complete suppression of $PbI_2$ emission within the ring. No increase in luminescence due to partially degraded perovskite phases or to $PbI_2$ band-edge emission was observed. These results highlight the ubiquity of photo-induced ion migration processes in OIHPs and their potential impact on the material quality and device performance. Our work also demonstrated that multi-modal optoelectronic spectroscopies that combine laser and converged electron-beam excitations in situ with variable PE energies and interaction volumes can be a valuable tool to study microscopic ion transport in emerging materials.

## Methods

### Sample fabrication

This section presents the method of fabrication for the samples considered in this study. The FTO-coated glass substrates were washed with the detergent aqueous solution, deionized water, and 2-propanol in the ultrasonic bath for 15 min sequentially. Then the FTO substrate was treated with UV-ozone for 20 min just before the perovskite precursor deposition. To prepare the 1.5M $MAPbI_3$ precursor solution, equal molar ratios of methylammonium iodide (MAI, GreatCell Solar, 99.99%) and $PbI_2$ (TCI, 99.99%) were dissolved in dimethylformamide (DMF, Sigma-Aldrich, 99.8%) and dimethyl sulfoxide (DMSO, Sigma-Aldrich, 99.8%) with a volume ratio of 4:1. To prepare the 1.4M $FAPbI_3$ precursor solution, equal molar ratios of formanidium iodide (FAI, GreatCell Solar, 99.99%), and $PbI_2$ were dissolved in DMF and DMSO with a volume ratio of 8:1. The perovskite precursor solution was spin-coated onto the FTO substrates at 4000 rpm for 30 s, and 0.6 ml of

diethyl ether was dripped onto the substrate at the time of 15 s to the end. The as-spun films were annealed at 100 °C for 10 min and 150 °C for 20 min for $MAPbI_3$ and $FAPbI_3$, respectively.

### SEM and EDS measurement

Secondary electron imaging and EDS measurements reported in this work was conducted inside a ThermoFisher Quanta 650 FEG SEM hosted at the University of California Santa Barbara. The secondary electrons were detected using a standard Everhart-Thornley detector, while the EDS spectra were collected using a ThermoFisher UltraDry 30 EDS detector and interpreted using the ThermoFisher Pathfinder software. A high vacuum of $1 \times 10^{-6}$ torr was maintained inside the SEM chamber for all measurements. The laser beam was generated from a Yb-doped-fiber laser (Clark-MXR IMPULSE) with a fundamental wavelength of 1030 nm, an average pulse width of 150 fs and a repetition rate of 5 MHz. The fundamental wavelength was converted to 515 nm using a BBO crystal and used as the source for light exposure. The laser beam was fed into the SEM chamber through a transparent viewport on the chamber wall.

A fresh area on the sample was used for every light exposure test. After the sample was exposed to light with a controlled intensity for a given amount of time, a secondary electron image was taken with an accelerate voltage of 5 keV and a beam current of 300 pA. A low beam current was used to minimize electron-beam-induced damage to the sample. A dwell time of 1 μs at each pixel was used to form the image. Then EDS line scans across the center of the light-exposure area at every ~7 μm were taken with a beam current of 300 pA. 100 linescans (100 ms per scan) were taken and averaged to minimize electron-beam induced damage, while enhancing the signal-to-noise ratio. The change in the absolute count of X-ray photons emitted from the exposed area was calculated as compared to the pristine background area.

### Cathodoluminescence

Cathodoluminescence (CL) measurements were performed with a Delmic Sparc CL collection module in a ThermoFisher Quattro environmental SEM hosted in the Center for Nanophase Materials Sciences (CNMS) at the Oak Ridge National Laboratory. A parabolic mirror was used to collect CL generated under electron-beam excitation and to deliver the laser to the sample. A retractable mirror was inserted to direct free-space coupled laser sources to the parabolic mirror and retracted to allow for in situ CL characterization using an Andor Kymera spectrograph. A 495-nm pulsed laser source generated from the second harmonic of a Mai Tai Ti:Sapphire laser with 100 fs pulse duration and 80 MHz repetition rate was focused to a spot size of 5 μm by the parabolic mirror. The optical penetration depth of the 495-nm laser in both $MAPbI_3$ and $FAPbI_3$ is about 70 nm[3]. A 532-nm Cobolt CW laser source was used in lieu of the pulsed laser for reference experiments included in the Supplementary Information. After variable laser exposure times with the retractable mirror inserted, the mirror was retracted, and CL spectrum images were acquired with the electron-beam conditions described in the manuscript. Because the SEM and the adjacent optics table are independently floated, an Aligna active beam stabilization system was used to dynamically correct for any movement of the SEM relative to the optics table. Blind NMF performed with scikit-learn[72] was used to track changes in the CL spectrum images as a function of laser exposure time.

## Data availability

The authors declare that all data supporting the findings of this study are available within the paper and its Supplementary information files or available from the corresponding authors upon request.

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

## Acknowledgements

The work conducted at University of California Santa Barbara (SEI and EDS) was based on research supported by US Department of Energy, Office of Basic Energy Sciences, under the award number DE-SC0019244 (for the development of the laser-coupled SEM) and by the US Army Research Office under the award number W911NF-19-1-0060 (for studying photo-induced physics in emerging materials). T.K. also acknowledges the support of the new faculty startup fund from Seoul National University and the Institute of Advanced Machines and Design at Seoul National University (SNU-IAMD). Cathodoluminescence microscopies were supported by the Center for Nanophase Materials Sciences (CNMS), which is a US Department of Energy, Office of Science User Facility at Oak Ridge National Laboratory. The sample fabrication at National Renewable Energy Laboratory (NREL) was supported by the US Department of Energy under Contract No. DE-AC36-08GO28308 with Alliance for Sustainable Energy, Limited Liability Company (LLC), the Manager and Operator of NREL. The authors at NREL also acknowledge the support on perovskite sample preparation from DE-FOA-0002064 and DE-EE0008790, funded by the US Department of Energy, Office of Energy Efficiency and Renewable Energy, Solar Energy Technologies Office. The views expressed in the article do not necessarily represent the views of the DOE or the U.S. Government.

## Author contributions

B.Liao, K.Z. and B.Lawrie conceived and supervised the project. S.P., Q.J. and K.Z. prepared the samples. T.K. and B.S. conducted the SEI, EDS and XPS measurements at UCSB. U.C. and R.G. contributed to the instrument development at UCSB. V.I., K.K. and B.Lawrie conducted SEI, CL and AFM measurements at ORNL. G.E. and B.Lawrie conducted the CL data analysis. T.K., B.Liao and B.Lawrie drafted the manuscript. All authors have commented and edited the manuscript.

## Competing interests

The authors declare no competing interests.
