## [Peer Review File · Nature Communications]

Mapping the Pathways of Photo-induced Ion Migration in Organic-inorganic Hybrid Halide PerovskitesREVIEWER COMMENTS

Reviewer #1 (Remarks to the Author):

The manuscript "Mapping the Pathways of Photo-induced Ion Migration in Organic-inorganic Hybrid Halide Perovskites" interrogates ion migration in FAPbI₃ and MAPbI₃ thin films by using electron microscopy (secondary electron images and EDS) with laser excitation of the perovskite. The authors were able to directly show which ions move in the perovskite using elemental mapping. They also used cathodoluminescence imaging to show the resulting changes in emission. This manuscript is an important study for understanding nanoscale dynamics in perovskite thin films. We need more studies like this to understand how ions move in perovskite thin films.

Overall, I think that this is an important paper, but could be improved by addressing the major and minor comments below regarding reversibility of these effects, light-induced damage, and the detection of Si and Sn with EDS. I also suggest that the authors perform a more exhaustive literature search. A quick google scholar search yielded important complementary results using cathodoluminescence and other methods that should be cited here.

Major comments:

Reversibility of light-induced effects - How many of these changes observed in the MAPI and FAPI films are reversible? Does the bright contrast in the secondary electron image fade away over time?

For the Si and Sn results, can the authors confirm that the Sn and Si are actually migrating? Perhaps the density of the perovskite regions or light-induced structural changes are creating gaps in the perovskite film, which is allowing the electron beam to excite more of the substrate.

How much does light exposure change the perovskite morphology? Can the authors take either SEM or AFM images before and after light exposure to see how much the laser light impacts the perovskite microstructure. It looks like it definitely changes the microstructure in Figure 5 A-F (as the author mentions), but what about in the first part of the study?

Minor comments:

Figure 1: What are the gray lines in C and D? The “SnO₂ (FTO)” is hard to read in part D.

Figure 2: Can the authors add a color bar to Figure 2? It is hard to tell quantitatively what the secondary electron enhanced contrast is. The image also looks very saturated at the highest brightness points.

Figure 5: Can the authors use a more informative color bar for the CL images? Many pixels look like they are saturated. The color bar in Figure S9 is much better.

Page 13: If easily-obtainable, can the authors take a PL image of an electron beam exposed area before and after exposure to see if the PL counts change? This would tell us if the electron beam is perturbing the properties of the perovskite film.

Page 13: Can the authors expand on the fact that “environmental degradation limited CL characterization”? I do not understand what this means.

Figure S10: Can you label the axes for the components?

There are many relevant references missing from this publication:

deQuilettes et al. observes similar ion migration using ToF-SIMS:
<https://www.nature.com/articles/ncomms11683>

Bischak et al. had a nearly identical optical setup to measure cathodoluminescence of photoinduced phase separation: <https://pubs.acs.org/doi/abs/10.1021/acs.nanolett.6b04453>,
<https://pubs.acs.org/doi/10.1021/acs.jpcclett.8b01512>

Reviewer #2 (Remarks to the Author):

Liao and co-authors reported the characterization of photo-induced ion migration in metal halide perovskites by coupling the in-situ laser illumination inside a SEM equipped with secondary electron imaging, energy-dispersive X-ray spectroscopy and cathodoluminescence with varying primary electron energies. This new methodology can help reveal the ion migration pathways in both the vertical and

horizontal directions to provide new insights for the photo-induced ion migration in perovskites. However, there are certain technical and scientific points that need to be carefully addressed before this paper can be considered for publication in Nature Communications.

1. In figure 1B, can authors explain which chemical element can be assigned to the peak at ~ 3.6 eV and ~ 4.2 eV, respectively?
2. How about the film quality of pristine MAPbI₃/FAPbI₃ perovskites? It seems the element ratio of Pb/I is significantly deviated from the formula 1:3 according to the EDS spectra. It is important to make sure that the samples have good quality before drawing conclusions. More characterizations of the initial perovskite films, such as film morphology, crystallinity, and PL are needed.
3. In page 8, the authors claimed "However, the iodine concentration profile is less consistent with the bright ring contrast observed in FAPbI₃." Please clarify this part because less consistency between iodine concentration profile and ring contrast was also observed in MAPbI₃.
4. In Figure 2, why some cracks are generated after laser illumination and then part of the cracks disappeared? Are the SEM images in Figure 2A or 2B are captured from the same area in the same sample?
5. In Figure 3B, can authors explain why the fitted radius of the iodine deficiency distribution in FAPbI₃ increased after being illuminated for 60 min?
6. In CL mapping, the authors claimed that the laser can induce 90K increase in the temperature of perovskite sample. How to isolate such a high temperature induced ion migration from the optical exposure counterpart?
7. What is the penetration depth of the 495-nm laser used for CL mapping?
8. Please clearly indicate the detected CL peaks in third and fourth row in Figure 5 instead of using different frame colors for the images.
9. Can authors include the CL mapping data collected at ~ 520 - 540 nm (i.e., from PbI₂) into Supporting information.
10. In Page 10, why does the perovskite luminescence of environmentally degraded FAPbI₃ sample can be partially recovered after laser irradiation? More evidence and explanations should be provided here to support the conclusion. And what are the conditions for environmental aging of the samples?
11. The authors have conducted the CL analysis of 97% FAPbI₃/3% MAPbBr₃ perovskite to explain the increased band-edge CL intensity. Is it possible to identify the phase separation behaviors in mixed halide perovskites (with high Br/I ratio) by using this tool? Although wide-bandgap perovskites have great potential in tandem solar cells, their phase stability under illumination is still a big concern. It would be more appealing if the authors could provide some experimental results and insights on this topic.
12. Please well-organize the figures and note in order in Supporting information, i.e., from Figure S1 to Figure SX. The current form and organization are confusing.
13. Please include the scale of wavelength of CL in Figure S10.

Reviewer #3 (Remarks to the Author):

Ion migration is a common but critical phenomenon in perovskite research, which directly influences device stability and performance. This manuscript by Kim and co-workers probes the pathway of laser-induced ion migration. By mapping the ion distribution along both lateral and vertical directions, authors claim that iodine and lead ions could migrate under illumination and bring out the migration pathway of different ions. However, since laser illumination also can cause perovskite decomposition, the conclusions should be more careful before excluding the decomposition influence. Based on my understanding, the phenomenon shown in this manuscript should be more related to laser-induced decomposition than ion migration. Thus, I cannot recommend publishing this manuscript in Nature Communications unless the authors provide more strong evidence the process is ion migration.

1. In this manuscript, the laser spot ($d=50\ \mu\text{m}$ on page 5) is much larger than the grain size ($<1\ \mu\text{m}$ in figure 5) of the perovskite thin film. In general, the grain boundary is a key factor that might accelerate the ion migration process. (Energy Environ. Sci., 2016, 9, 1752-1759) This part discussion is missing in this manuscript. To exclude the grain boundary influence, perovskite single crystal without grain boundaries might be a better choice than polycrystalline thin films.

2. On page 4, the 2nd line from the bottom, the authors mention the sample composition as 'MAPbI₃, FAPbI₃, and their hybrid'. The hybrid sample composition they use after is (FAPbI₃)_{0.97}(MAPbBr₃)_{0.03}, which contains Br, not simply MAPbI₃ and FAPbI₃ hybrid. I would suggest using the real composition rather than 'their hybrid'.

3. In figure 2, it is clear that the contrast of SE images significantly changes by illumination. However, this also can be caused by decomposition. The morphology before and after illumination and the morphology influence is needed.

4. To evaluate the vertical direction ion distribution, the authors used a 30 keV PE, which can get through the whole perovskite layer. I am not convincing that the Sn and Si signal in figure 3 is caused by ion migration rather than by substrate because authors claimed the 30 keV PE could probe the whole sample, including the substrate. A cross-section EDS elemental analysis directly shows the vertical direction of ion distribution.

5. On page 7, the 3rd line from the bottom, authors claim 'likely due to a larger bandgap in MAPbI₃ and, thus a weaker optical absorption.' However, the absorption coefficients of MAPbI₃ and FAPbI₃ are similar at the laser wavelength (at 2.4 eV). This is not a proper explanation for the spot size difference.

6. On page 12, line 3 of section C, the authors mention 'However, no photoluminescence near the bandgap of Pbl₂ was observed.' This cannot support authors' opinion, because the PL of Pbl₂ is too weak to measure at room temperature (J. Mater. Sci., 2008, 43, 525-529).

7. On page 13, the authors claim that 'environmental degradation limited CL characterization of the pure FAPbI₃ and MAPbI₃ films.' Considering the measurement is under a high vacuum, environmental

degradation should not be a reason that cannot measure CL. And also, the CL spectra in figure S10 do not have x-axis.

8. The explanations of two peaks in CL spectra are missing in this manuscript. What is the reason for the two peaks? Phase segregation or something else? If it is caused by phase segregation, why does the pristine sample also has two peaks? Is there any reason for the pristine sample just having less than 300 intensities in the CL spectrum?

Revision Report NCOMMS-22-41216 #1

Mapping the Pathways of Photo-induced Ion Migration in Organic-inorganic Hybrid Halide Perovskites

T. Kim et al.

We appreciate the reviewers' constructive criticism and valuable perspective on the technical details and overall message of our paper, which have helped us think more thoroughly about our results and improve the quality of the manuscript. In particular, we conducted a series of new supplementary measurements, including techniques such as AFM and XPS and we also repeated *in situ* characterization of new samples under a wider variety of laser powers to clarify our previous findings. We added two authors (B. Shaheen and K. Kelley) to this work, who conducted some of the additional experiments. As a result, we extensively revised both the main text (the entire section on cathodoluminescence measurements) and the supplementary information, including 10 new figures added to the supplementary information. In the following, we address the reviewers' comments point by point and explain our response and resulting actions in the manuscript. The reviewers' comments are included in italic for clarity, while our responses are presented in regular font and highlighted in blue.

Additional Changes:

After performing additional cathodoluminescence microscopies on new samples at various laser powers, we were able to separate photoinduced effects that are present at lower laser powers and photothermal effects that are present at higher laser powers. The updated cathodoluminescence results are highlighted in the updated Figure 5 and in Supplemental Figures S16-S20.

Response to Reviewer 1:

Comments: *The manuscript "Mapping the Pathways of Photo-induced Ion Migration in Organic-inorganic Hybrid Halide Perovskites" interrogates ion migration in FAPbI₃ and MAPbI₃ thin films by using electron microscopy (secondary electron images and EDS) with laser excitation of the perovskite. The authors were able to directly show which ions move in the perovskite using elemental mapping. They also used cathodoluminescence imaging to show the resulting changes in emission. This manuscript is an important study for understanding nanoscale dynamics in perovskite thin films. We need more studies like this to understand how ions move in perovskite thin films.*

Overall, I think that this is an important paper, but could be improved by addressing the major and minor comments below regarding reversibility of these effects, light-induced damage, and the detection of Si and Sn with EDS. I also suggest that the authors perform a more exhaustive literature search. A quick google scholar search yielded important complementary results using cathodoluminescence and other methods that should be cited here.

Response: We thank the reviewer for the generally positive evaluations of our manuscript. We address the comments below.

Comments: *Major comments:*

1. *Reversibility of light-induced effects - How many of these changes observed in the MAPI and FAPI films are reversible? Does the bright contrast in the secondary electron image fade away over time?*

Response: Due to the iodine deficiency near the surface that can be caused by I_2 escaping to the vacuum, we do not expect the observed ion migration effects to be reversible. To confirm it, we conducted additional tests by monitoring the change in the light-induced secondary electron contrast over time. The results are shown here in Fig. R1. Figure R1A shows the secondary-electron contrast in FAPbI₃ induced by light exposure (5 minutes, fluence: $106.6 \mu\text{J cm}^{-2}$). Then the laser was turned off and the sample was stored inside the SEM vacuum chamber (vacuum level: 10^{-6} torr) up to 12 hours. The secondary-electron contrast after 4 hours, 10 hours and 12 hours are shown in Fig. R1B-D, indicating the contrast remains the same within this period of time and the photo-induced change is irreversible. We added this figure to the Supplementary Information (Section I.E) and added relevant discussion to the revised Main Text, near the discussion of secondary electron images in Fig. 2.

We attempted to perform repeated CL microscopy of the same areas to strengthen this point, but we found that, while the electron-beam doses used for CL microscopy did not introduce any obvious damage within an individual scan, subsequent scans exhibited moderate reductions in brightness as a result of the composite electron-beam induced damage introduced during the initial CL microscopy.

Figure R1. Reversibility test of the photo-induced secondary electron contrast in FAPbI₃. A: The photo-induced secondary electron contrast in FAPbI₃ after light exposure for 5 minutes (fluence: $106.6 \mu\text{J cm}^{-2}$). B-D: the secondary-electron contrast after storing the sample in the vacuum chamber for 4 hours, 10 hours and 12 hours, respectively. The contrast induced by 5-minutes light exposure remains after 12 hours, suggesting the effect is not reversible spontaneously.

Comments: 2. For the Si and Sn results, can the authors confirm that the Sn and Si are actually migrating? Perhaps the density of the perovskite regions or light-induced structural changes are creating gaps in the perovskite film, which is allowing the electron beam to excite more of the substrate.

Response: We thank the reviewer for pointing out a possibility that would also lead to increased Si and Sn X-ray counts in the EDS mapping after light exposure. To investigate this possibility, we conducted a comparative study of a FAPI sample deposited on a 80-nm gold-coated FTO glass substrate. If indeed the density of the perovskite is changed after exposure or light-induced structural changes create gaps in the film so that the electron beam can excite the substrate more, we would expect an increased X-ray counts associated with the gold characteristic line. The results are shown here in Fig. R2, where the changes in the X-ray counts normalized to the background values associated with silicon (K_{α} band, 1.74 keV), tin (L_{α} band, 3.44 keV for 30 keV PE and M band, 0.69 keV for 5 keV PE) and gold (M band, 2.12 keV) after light exposure (fluence: $105 \mu\text{J cm}^{-2}$, 10 minutes exposure) are displayed. The shaded area denotes the exposed region in the sample. First of all, the X-ray counts associated with gold were minimally changed within the background fluctuation after the light exposure. The X-ray counts associated with tin showed a similar behavior. These observations suggest that it is unlikely that the perovskite film passes more electrons to the substrate after light exposure. Surprisingly, we noticed the X-ray counts associated with silicon when probed by 30 keV PEs were still significantly enhanced in this case. While we do not expect silicon ions to migrate across the FTO and gold layers, the observed enhancement likely suggests an enrichment of silicon in FTO and even gold within the exposed region. Unfortunately, at this stage we do not have access to characterization techniques within a reasonable amount of time that can directly prove the presence of silicon and/or tin ions inside perovskite after light exposure. To clarify this point, we added the new test results into the Supplementary Information (Section I.G) and a detailed discussion in the revised Main Text, near the discussion of EDS elemental analysis in Fig. 3.

Figure R2. EDS mapping of a FAPbI_3 -on-gold sample after light exposure. The sample was exposed to light with a fluence of $105 \mu\text{J cm}^{-2}$ for 10 minutes. The left, center and right panels show the changes in the X-ray counts normalized to the background values associated with silicon (K_{α} band, 1.74 keV), tin (L_{α} band, 3.44 keV for 30 keV PE and M band, 0.69 keV for 5 keV PE) and gold (M band, 2.12 keV) after light exposure. The shaded area denotes the exposed region in the sample.

Comments: 3. How much does light exposure change the perovskite morphology? Can the authors take either SEM or AFM images before and after light exposure to see how much the laser light impacts the perovskite microstructure. It looks like it definitely changes the microstructure

in Figure 5A-F (as the author mentions), but what about in the first part of the study?

Response: As suggested by the reviewer, we took high-resolution SEM and AFM images after the light exposure of FAPbI₃ to examine the microstructural change after light exposure. A representative SEM image is shown here in Fig. R3 and an AFM image is shown in Fig. R4. Consistent with the observations shown in Fig. 5 in the main text, the area exposed to light has smaller grains compared to the pristine area according to the SEM and AFM images. The bright “ring”-shaped edge area has a width of a few micrometers and is likely due to an increased electrical resistance that leads to a local accumulation of charge. These images are now also added to the Supplementary Information and we added relevant discussion to the main text as well, near the discussion of diffusion process.

Figure R3. High-resolution SEM images of FAPbI₃ after light exposure, showing the microstructural change. The sample was exposed to light with a fluence of $105 \mu\text{J cm}^{-2}$ for 30 minutes. 5-keV PEs were used to generate the images. It is clearly shown that light-exposed area has smaller grains compared to the pristine area. The bright “ring”-shaped edge of the exposure area is most likely due to an increased electrical resistance that leads to local charge accumulation.

Comments: *Minor comments:*

Figure 1: What are the gray lines in C and D? The “SnO₂ (FTO)” is hard to read in part D.

Response: The gray lines in Figs. 1C and D indicate the interfaces between different materials inside the specimen. In the revised Fig. 1, the interfaces are annotated at the top of each figure. Additionally, the font sizes and colors of the labels in the inset of Fig. 1D are changed to improve the visibility.

Comments: Figure 2: Can the authors add a color bar to Figure 2? It is hard to tell quantitatively what the secondary electron enhanced contrast is. The image also looks very saturated at the highest brightness points.

Response: We thank the reviewer for the suggestion. We did not add color bars to the SEM images in Fig. 2 since the contrast of these images can only be interpreted qualitatively, not quantitatively. This is because the contrast observed in SEM images is not linearly proportional to the number of secondary electrons detected by the ETD, due to nonlinear gains in the photomultiplier and the amplifier and the image acquisition and

Figure R4. AFM morphology scan of FAPbI₃ after light exposure, showing the microstructural change. The sample was exposed to light with a fluence of 105 $\mu\text{J cm}^{-2}$ for 30 minutes. The light exposure area is denoted by the blue dashed circle. The AFM image shows smaller grains within the light-exposed area.

processing software used. Therefore, in this case, adding a quantitative color bar may actually lead to over-interpretation of the SEM images.

Comments: Figure 5: Can the authors use a more informative color bar for the CL images? Many pixels look like they are saturated. The color bar in Figure S9 is much better.

Response: In response to concerns raised by each of the reviewers, we fabricated fresh FAPbI₃, MAPbI₃, and 97% FAPbI₃/3% MAPbBr₃ films and measured the cathodoluminescence of each of these films under various laser powers. These new results are discussed in greater detail below, but the updated Figure 5 now includes CL analysis of a MAPbI₃ film irradiated with a 100- μW , 495-nm pulsed laser source using a color scheme consistent with those used in the Supporting Information.

Comments: Page 13: If easily-obtainable, can the authors take a PL image of an electron beam exposed area before and after exposure to see if the PL counts change? This would tell us if the electron beam is perturbing the properties of the perovskite film.

Response: First of all, we want to emphasize that the experiments described in the manuscript include two fundamentally different scales of electron-beam exposure. For the *in situ* EDS experiments, the electron-beam exposure of the perovskite samples was minimal. The electron beam was blocked when the sample was being exposed to light. Then one SEM image or EDS scan was done after light exposure with minimal electron dose. A new location on the sample was used for every light exposure experiment. A similar approach was used for the *in situ* CL microscopy: the electron-beam was blocked during laser exposure, and the SEM image and CL were then acquired concurrently, but with a greatly increased dose as a result of the 100 ms exposure time per pixel required for the CL spectra.

We struggled to measure a PL image of the exposed area. While we can measure PL in situ in the SEM, the laser spot size would be the same as the laser spot size used for our CL experiments, so we would not have the necessary spatial resolution to perform any mapping. Similarly, ex situ PL microscopy proved challenging because we were constrained to PL in ambient conditions and time constraints made it challenging to locate the irradiated areas after removal from the SEM before environmental degradation became a concern. However, we were able to take supporting AFM images that are discussed in greater detail below.

More importantly, we could also use repeated CL microscopy to probe the effects of electron-beam induced damage. While no obvious electron-beam induced damage is visible in any of the CL images, repeated CL microscopy of the same areas resulted in the observation of up to 50% attenuation of the excitonic CL. In contrast, after many SEM images were acquired of the same area using dwell times of up to 10 μ s per pixel, no obvious damage was observed in subsequent CL images. So while no measurable electron-beam induced damage was present in the EDS data, the reader should be conscious of moderate electron-beam induced damage distributed uniformly across the CL data. We have added a discussion of this concern to the text describing Figure 5.

Comments: Page 13: Can the authors expand on the fact that “environmental degradation limited CL characterization”? I do not understand what this means.

Response: The MAPbI₃ and FAPbI₃ samples used in the original CL experiments were degraded after shipment due to ambient environmental exposure over a period of several weeks, so we focused on the 97% FAPbI₃/3% MAPbBr₃ films, which exhibited no signs of degradation. In the revised version of the manuscript and Supporting Information, we repeated all of these measurements across a broader range of laser powers for pristine samples.

Comments: Figure S10: Can you label the axes for the components?

Response: As mentioned above, we have added several additional CL spectrum images to the Supporting Information, though we did use a similar format to that used in Fig. S10 in order to minimize excessive whitespace in the figures. Note that the horizontal field width for all of the images is listed in the caption.

Comments: There are many relevant references missing from this publication: deQuilettes et al. observes similar ion migration using ToF-SIMS:

<https://www.nature.com/articles/ncomms11683>

Bischak et al. had a nearly identical optical setup to measure cathodoluminescence of photoinduced phase separation:

<https://pubs.acs.org/doi/abs/10.1021/acs.nanolett.6b04453>,

<https://pubs.acs.org/doi/10.1021/acs.jpcllett.8b01512>

Response: We thank the reviewer for pointing out these important references. We have now added these references to the revised manuscript and added proper discussions. Indeed, there is rich literature about ion migrations in perovskites and we tried to include as many most relevant references to our work as possible while maintaining a manageable size of the reference list.

Response to Reviewer 2:

Comments: *Liao and co-authors reported the characterization of photo-induced ion migration in metal halide perovskites by coupling the in-situ laser illumination inside a SEM equipped with secondary electron imaging, energy-dispersive X-ray spectroscopy and cathodoluminescence with varying primary electron energies. This new methodology can help reveal the ion migration pathways in both the vertical and horizontal directions to provide new insights for the photo-induced ion migration in perovskites. However, there are certain technical and scientific points that need to be carefully addressed before this paper can be considered for publication in Nature Communications.*

Response: We are grateful to the reviewer for his/her constructive comments.

Comments: 1. *In Figure 1B, can authors explain which chemical element can be assigned to the peak at 3.6 keV and 4.2 keV, respectively?*

Response: According to the characteristic X-ray database, the peak at ~ 3.6 keV is primarily due to the tin L_{β_1} line (3.66 keV), while the peak at ~ 4.2 keV can be attributed to the tin L_{γ_1} line, 4.13 keV) and the iodine L_{β_1} line (4.22 keV). We have labeled these peaks in Fig. 1 in the revised manuscript.

Comments: 2. *How about the film quality of pristine MAPbI₃/FAPbI₃ perovskites? It seems the element ratio of Pb/I is significantly deviated from the formulate 1:3 according to the EDS spectra. It is important to make sure that the samples have good quality before drawing conclusions. More characterizations of the initial perovskite films, such as film morphology, crystallinity, and PL are needed.*

Response: We want to point out that the EDS spectra shown in the manuscript were based on the absolute X-ray photon counts and they do not directly reflect the relative atomic ratio. A sophisticated conversion model based on the elemental X-ray emission cross section and X-ray transport process needs to be used to convert the absolute X-ray counts into relative atomic ratio, which can depend on the data processing software and is not quantitatively reliable. Instead, we conducted X-ray photoelectron spectroscopy (XPS) characterization of our FAPbI₃ and MAPbI₃ samples to determine the relative atomic ratio. XPS detects photoelectrons from the very surface of the sample and thus does not rely on modeling the transport process inside the sample. The measured electron binding energy corresponding to Pb 4f and I 3d levels are shown in Fig. R5. Based on the XPS spectra averaged over several locations on the samples, the iodine-to-lead ratio is estimated to be 4.1 in FAPbI₃ and 3.3 in MAPbI₃. The small deviation from the nominal ratio of 3 might be due to slight sample surface degradation during sample shipping from NREL to UCSB, which is consistent with a previous study [1]. We further provide the X-ray diffraction data in Fig. R6 and the PL data in Fig. R7 that confirm the high quality of the samples. The SEM morphological data is shown in Fig. R3. Zhu group at NREL has extensive experience of synthesizing high-quality perovskite thin films and systematic film-quality characterization data has been reported in Zhu group's previous publications (e.g. [2, 3]). The same procedure as used in these previous publications was followed when fabricating the samples for this work. The additional characterization data is now added to the Supplementary Information.

Comments: 3. *In page 8, the authors claimed "However, the iodine concentration profile is less consistent with the bright ring contrast observed in FAPbI₃." Please clarify this part because less*

Figure R5. High-resolution XPS spectra of I 3d and Pb 4f levels in perovskite thin films. The top panels are the spectra for FAPbI₃ and the bottom panels are the spectra for MAPbI₃.

consistence between iodine concentration profile and ring contrast was also observed in MAPbI₃.

Response: Here the purpose of the quoted statement is to emphasize that the measured iodine concentration profile is more like a Gaussian, while the observed SE contrast has a bright ring shape in FAPbI₃. In contrast, the light-induced contrast in MAPbI₃ has a Gaussian shape (see Fig. 2B) rather than the ring shape observed in FAPbI₃. To clarify, we modified the quoted sentence to “However, the iodine concentration profile within the area illuminated by light showed a Gaussian shape, which does not explain the bright ring-shaped contrast observed in FAPbI₃” in the revised version.

Comments: 4. In Figure 2, why some cracks are generated after laser illumination and then part of the cracks disappeared? Are the SEM images in Figure 2A or 2B are captured from the same area in the same sample?

Response: To minimize sample damage due to the electron beam exposure, we conducted each light-exposure experiment at a new location on the sample. Namely, each SEM image shown in Fig. 2 was taken at a different location on the sample. The cracks shown in some of the SEM images in Fig. 2 are intrinsic defects on the sample surface and not created by light exposure. To clarify this point, we added “To minimize the potential sample damage due to the electron beam exposure, a new location on the sample was chosen for each of the light-exposure experiments reported here” to the main text and the caption to Fig. 2.

Comments: 5. In Figure 3B, can authors explain why the fitted radius of the iodine deficiency distribution in FAPbI₃ increased after being illuminated for 60 min?

Figure R6. XRD scans of our MAPbI₃ and FAPbI₃ samples. The stars label the peaks associated with the FTO substrate.

Response: The fitted radius of the iodine deficiency distribution increased with time continuously as a result of light-driven migration of iodine ions. We extracted the diffusion coefficient from the time dependence of the fitted radius, as shown in Fig. 3B.

Comments: 6. In CL mapping, the authors claimed that the laser can induce 90K increase in the temperature of perovskite sample. How to isolate such a high temperature induced ion migration from the optical exposure counterpart?

Response: We agree with the reviewer that such a high temperature rise make it difficult to isolate thermally driven and optically driven effects. In the revised manuscript, we conducted new CL measurements with a lower optical power to ensure the temperature rise is lower than 30 K. The new results are reported in the updated Fig. 5, and they exhibit a substantial contrast compared with the higher laser power data that is now included in the Supporting Information.

Comments: 7. What is the penetration depth of the 495-nm laser used for CL mapping?

Response: According to the literature (e.g. Fig. 13e in [4]), the optical penetration depth of the 495-nm laser in both MAPbI₃ and FAPbI₃ is about 70 nm. We have added this information to the Methods section.

Comments: 8. Please clearly indicate the detected CL peaks in third and fourth row in Figure 5 instead of using different frame colors for the images.

Response: The excitonic CL cannot be fit to a single Gaussian or Lorentzian peak because a partially degraded metastable state typically appears even in very high quality perovskite films. We use the NMF decomposition to delineate the partially degraded state from the pure excitonic state, but because the two states are nearly resonant with one another, we chose to identify them by labeling components 1, 2, and 3 in the second column with the intensity maps associated with those columns illustrated in the following columns. We hope this representation is appropriately clear.

Comments: 9. Can authors include the CL mapping data collected at 520-540 nm (i.e., from PbI₂) into Supporting information.

Response: While no CL was observed from the PbI₂ band in the original data, we did

Figure R7. Photoluminescence Characterization of Our MAPbI₃ and FAPbI₃ samples. A continuous 532-nm excitation laser was used with 2 μ W power and 200 ms collection time.

observe isolated low intensity PbI₂ CL in the fresh MAPbI₃ films that are illustrated in the revised Fig. 5. Because the PbI₂ CL is very weak compared with the excitonic CL, the NMF reconstruction did not isolate it as well as we had hoped, so we also included raw bandpass filtered PbI₂ CL data in the final column of the revised Fig. 5.

Comments: 10. In Page 10, why does the perovskite luminescence of environmentally degraded FAPbI₃ sample can be partially recovered after laser irradiation? More evidence and explanations should be provided here to support the conclusion. And what are the conditions for environmental aging of the samples?

Response: The FAPbI₃ and MAPbI₃ samples were originally stored in ambient conditions for a period of several weeks while the hybrid sample was under vacuum in the SEM. In the updated experiments, the samples were shipped in a vacuum sealed container and loaded together into the SEM immediately after receipt. All subsequent experiments were completed within a one week period without breaking vacuum in the SEM. The enhanced perovskite luminescence reported in the original manuscript was only present for higher laser powers. At the reduced powers used in the new measurements, no such enhancement was observed. This suggests that the partial recovery and luminescence enhancement were a result of photothermal effects and not directly related to the photoinduced ion migration effects described in the manuscript. However, a more detailed understanding of those photothermal interactions is beyond the scope of this manuscript.

Comments: 11. The authors have conducted the CL analysis of 97% FAPbI₃/3% MAPbBr₃ perovskite to explain the increased band-edge CL intensity. Is it possible to identify the phase separation behaviors in mixed halide perovskites (with high Br/I ratio) by using

this tool? Although wide-bandgap perovskites have great potential in tandem solar cells, their phase stability under illumination is still a big concern. It would be more appealing if the authors could provide some experimental results and insights on this topic.

Response: In the revised manuscript, we have moved the discussion of the 97% FAPbI₃/3% MAPbBr₃ to the Supporting Information in order to improve the focus on the FAPbI₃ and MAPbI₃ films in the manuscript. Unfortunately, we were not able to take any CL images of mixed halide perovskites with high Br/I ratios, though that may be possible in future work.

Comments: 12. Please well-organize the figures and note in order in Supporting information, i.e., from Figure S1 to Figure SX. The current form and organization are confusing.

Response: As suggested by the reviewer, we have reordered the figures and sections in the Supplementary Information following the order they are referenced in the Main Text.

Comments: 13. Please include the scale of wavelength of CL in Figure S10.

Response: It has been added in the revised version.

Response to Reviewer 3:

Comments: *Ion migration is a common but critical phenomenon in perovskite research, which directly influences device stability and performance. This manuscript by Kim and co-workers probes the pathway of laser-induced ion migration. By mapping the ion distribution along both lateral and vertical directions, authors claim that iodine and lead ions could migrate under illumination and bring out the migration pathway of different ions. However, since laser illumination also can cause perovskite decomposition, the conclusions should be more careful before excluding the decomposition influence. Based on my understanding, the phenomenon shown in this manuscript should be more related to laser-induced decomposition than ion migration. Thus, I cannot recommend publishing this manuscript in Nature Communications unless the authors provide more strong evidence the process is ion migration.*

Response: We thank the reviewer for pointing out the importance and potential impact of thermal decomposition on our findings. We provide a detailed analysis in the following.

Comments: 1. *In this manuscript, the laser spot ($d=50\ \mu\text{m}$ on page 5) is much larger than the grain size ($< 1\ \mu\text{m}$ in Figure 5) of the perovskite thin film. In general, the grain boundary is a key factor that might accelerate the ion migration process. (Energy Environ. Sci., 2016, 9, 1752-1759) This part discussion is missing in this manuscript. To exclude the grain boundary influence, perovskite single crystal without grain boundaries might be a better choice than polycrystalline thin films.*

Response: We agree with the reviewer that grain boundaries can play a significant role in facilitating ion migration observed in our study and we thank the reviewer for pointing out an important reference, which we have now cited in the revised manuscript with a discussion. Indeed single crystalline and polycrystalline crystal samples should show different ion migration properties due to the grain boundary contributions. As shown in a previous study of ion migration in single crystalline perovskites [5], the diffusion co-

efficient of halide ions in single crystalline perovskites is on the order of 10^{-13} to 10^{-12} $\text{cm}^2 \text{s}^{-1}$ even at an elevated temperature of 100°C , which is at least two orders of magnitude smaller than the value we measured in polycrystalline samples ($\sim 3.5 \times 10^{-10}$ $\text{cm}^2 \text{s}^{-1}$). This significant difference can be attributed to the contribution of the grain boundaries. We added this discussion to the revised manuscript. We agree with the reviewer that it would be beneficial to directly compare single crystalline and polycrystalline samples with the same measurement technique. Unfortunately, due to the low electrical conductivity and the large sample thickness of single crystalline perovskites (and the difficulty to thin down the single crystals without breaking them), we encountered severe sample charging issue when attempting the EDS mapping that led to unreliable results. On the other hand, we want to emphasize that the mainstream perovskite solar cells are currently based on polycrystalline thin films. Therefore, it is equally, if not more, important to understand ion migration in the presence of grain boundaries in polycrystalline thin films, rather than to exclude the contribution from grain boundaries.

Comments: 2. *On page 4, the 2nd line from the bottom, the authors mention the sample composition as “MAPbI₃, FAPbI₃, and their hybrid”. The hybrid sample composition they use after is (FAPbI₃)_{0.97}(MAPbBr₃)_{0.03}, which contains Br, not simply MAPbI₃ and FAPbI₃ hybrid. I would suggest using the real composition rather than “their hybrid”.*

Response: For consistency and simplicity of this work, we have removed the data and the discussion of the hybrid sample from the revised manuscript. Instead, we conducted *in situ* cathodoluminescence measurements of pure FAPbI₃ and MAPbI₃ samples and presented the data in the updated Fig. 5 in the revised manuscript. Discussion of the hybrid films was moved to the revised Supporting Information.

Comments: 3. *In Figure 2, it is clear that the contrast of SE images significantly changes by illumination. However, this also can be caused by decomposition. The morphology before and after illumination and the morphology influence is needed.*

Response: Indeed, thermal decomposition of hybrid perovskites has been reported in many previous studies of their degradation mechanisms. In this work, we took great care to minimize the contribution of thermal decomposition by limiting the optical fluence used to very low levels to ensure the steady-state temperature rise is below 30 K. Furthermore, the complete suppression of PbI₂ CL in the irradiated areas, combined with the absence of any enhancement of partially degraded excitonic CL in any of the irradiated samples suggests that decomposition did not play an appreciable role in our findings (more discussion in our response to Comment 6 below). In addition, we want to note that ion migration and decomposition are intrinsically intertwined processes that cannot be distinguished in an absolute sense. Any decomposition process will lead to a nonuniform distribution of ions that will drive ion migration. In the mean time, any ion migration process can also lead to local changes of the material composition that can be termed “decomposition” if the change is significant. In our case, due to the lack of spectral signature of the decomposition product PbI₂, we believe the composition change in our samples due to ion migration was small. Accordingly, we prefer to term the effect “ion migration” rather than “decomposition”. As requested by the reviewer, we provide the SEM and AFM morphological data in Fig. R3 and Fig. R4, where light exposure is

shown to lead to smaller grain sizes compared to the pristine region.

Comments: 4. *To evaluate the vertical direction ion distribution, the authors used a 30 keV PE, which can get through the whole perovskite layer. I am not convinced that the Sn and Si signal in Figure 3 is caused by ion migration rather than by substrate because authors claimed the 30 keV PE could probe the whole sample, including the substrate. A cross-section EDS elemental analysis directly shows the vertical direction of ion distribution.*

Response: We agree with the reviewer that the observed increase of the Si and Sn signal can be due to more electrons arriving at and exciting the substrate after the light exposure of the perovskite layers. To rule out this possibility, we conducted a comparative study of a FAPbI₃-on-gold-coated-FTO sample, as described in detail in our response to Comment 2 of Reviewer 1. The result is shown in Fig. R2. If indeed the increased substrate excitation was the reason, we should see an increased X-ray signal associated with gold, which we did not observe. We also found that the tin signal in this case remained the same before and after light exposure, suggesting that the increased substrate excitation was unlikely the reason for the increased Si and Sn signal in our study. In principle, a cross-sectional EDS analysis would be ideal to directly prove the existence of Si and Sn inside the perovskite layer. However, it is challenging to create the cross section *in situ* right at the light exposed area. Furthermore, taking the sample out of the SEM chamber is unavoidable to fabricate the cross-sectional sample, which may lead to additional sample degradation that can complicate the analysis.

Comments: 5. *On page 7, the 3rd line from the bottom, authors claim “likely due to a larger bandgap in MAPbI₃ and, thus a weaker optical absorption.” However, the absorption coefficients of MAPbI₃ and FAPbI₃ are similar at the laser wavelength (at 2.4 eV). This is not a proper explanation for the spot size difference.*

Response: We agree with the reviewer that the absorption coefficients of MAPbI₃ and FAPbI₃ are similar at 2.4 eV despite the larger band gap of MAPbI₃. We have modified the quoted sentence to remove the misleading statement. The observed slower response in MAPbI₃ thus might be caused by a lower iodine diffusivity.

Comments: 6. *On page 12, line 3 of section C, the authors mention “However, no photoluminescence near the bandgap of PbI₂ was observed.” This cannot support authors’ opinion, because the PL of PbI₂ is too weak to measure at room temperature (J. Mater. Sci., 2008, 43, 525-529).*

Response: We thank the reviewer for bringing this literature to our attention. We want to point out that, in this literature, a 532-nm pump laser was used to excite the sample, which could not directly excite the PL in PbI₂ at 515 nm. In our CL measurement, however, the electron beam energy is high enough to efficiently generate luminescence at 515 nm from PbI₂. This is shown in Fig. 5 and Figs. S16, S19 and S21, where clear emission at 515 nm was observed. Therefore, we are confident that the CL measurement is a sensitive probe to the presence of PbI₂.

Comments: 7. *On page 13, the authors claim that “environmental degradation limited CL characterization of the pure FAPbI₃ and MAPbI₃ films.” Considering the measurement is under a high vacuum, environmental degradation should not be a reason that cannot measure CL. And also, the CL spectra in Figure S10 do not have x-axis.*

Response: As noted above, that environmental degradation occurred because the sam-

ples were not initially all loaded in the SEM at the same time. As suggested by the reviewer, we conducted new *in situ* CL measurements of pure FAPbI₃ and MAPbI₃ films and provided the updated result in Fig. 5 of the revised manuscript. We have also added the x-axis to Fig. S10.

Comments: 8. *The explanations of two peaks in CL spectra are missing in this manuscript. What is the reason for the two peaks? Phase segregation or something else? If it is caused by phase segregation, why does the pristine sample also has two peaks? Is there any reason for the pristine sample just having less than 300 intensities in the CL spectrum?*

Response: CL microscopies of perovskite films generally cannot be fit well to a single Gaussian or Lorentzian peak because of the presence of an intermediate phase that results from partial degradation of the film [6, 7]. Electron-beam induced decomposition can be minimized by minimizing the electron beam current and dwell time. In this case, we used a 100 ms dwell time and a 7 pA current to minimize the beam induced damage while still measuring a signal with reasonable signal-to-noise ratio. We have revised the discussion of the CL microscopy in the manuscript to address this concern.

References

- [1] Wei-Chun Lin, Wei-Chun Lo, Jun-Xian Li, Yi-Kai Wang, Jui-Fu Tang, and Zi-Yun Fong. In situ XPS investigation of the X-ray-triggered decomposition of perovskites in ultra-high vacuum condition. *npj Materials Degradation*, 5(1):13, 2021.
- [2] Zhi Guo, Yan Wan, Mengjin Yang, Jordan Snaider, Kai Zhu, and Libai Huang. Long-range hot-carrier transport in hybrid perovskites visualized by ultrafast microscopy. *Science*, 356(6333):59–62, 2017.
- [3] Jinhui Tong, Zhaoning Song, Dong Hoe Kim, Xihan Chen, Cong Chen, Axel F Palmstrom, Paul F Ndione, Matthew O Reese, Sean P Dunfield, Obadiah G Reid, et al. Carrier lifetimes of $\sim 1 \mu\text{s}$ in Sn-Pb perovskites enable efficient all-perovskite tandem solar cells. *Science*, 364(6439):475–479, 2019.
- [4] Jin Young Kim, Jin-Wook Lee, Hyun Suk Jung, Hyunjung Shin, and Nam-Gyu Park. High-efficiency perovskite solar cells. *Chemical Reviews*, 120(15):7867–7918, 2020.
- [5] Minliang Lai, Amael Obliger, Dylan Lu, Christopher S Kley, Connor G Bischak, Qiao Kong, Teng Lei, Letian Dou, Naomi S Ginsberg, David T Limmer, et al. Intrinsic anion diffusivity in lead halide perovskites is facilitated by a soft lattice. *Proceedings of the National Academy of Sciences*, 115(47):11929–11934, 2018.
- [6] Chuanxiao Xiao, Zhen Li, Harvey Guthrey, John Moseley, Ye Yang, Sarah Wozny, Helio Moutinho, Bobby To, Joseph J Berry, Brian Gorman, et al. Mechanisms of electron-beam-induced damage in perovskite thin films revealed by cathodoluminescence spectroscopy. *The Journal of Physical Chemistry C*, 119(48):26904–26911, 2015.
- [7] Ethan J Taylor, Vasudevan Iyer, Bibek S Dhami, Clay Klein, Benjamin J Lawrie, and Kannatassen Appavoo. Hyperspectral nanoscale mapping of hybrid perovskite photophysics at the single grain level. *arXiv preprint arXiv:2201.06546*, 2022.

REVIEWER COMMENTS

Reviewer #1 (Remarks to the Author):

I thank the authors for addressing my concerns. I think this is an important study and look forward to seeing it published.

Reviewer #2 (Remarks to the Author):

I am satisfied with the revised manuscript which has addressed my comments in the previous round. I recommend it to be accepted.

Reviewer #3 (Remarks to the Author):

Since the authors' responses didn't give more strong evidence to support their conclusion, I insist this is a decomposition process rather than ion migration, so I cannot recommend publishing this manuscript on Nature Communications.

1. The non-reversible changes in Fig. R1 clearly indicate this is a decomposition process. The authors also admitted this on page 11:

'Photo-induced decomposition of FAPbI_3 into PbI_2 and, further, into I_2 vapor leads to an iodine deficiency near the surface, which creates an iodine concentration gradient that drives the diffusion of iodine ions towards the surface.'

2. The laser fluence used in this manuscript is 106.6 uJ cm^{-2} , which refers to an excitation density of 2.77×10^{17} photons cm^{-2} per pulse, average excitation density of 1.385×10^{24} photons $\text{cm}^{-2} \text{ s}^{-1}$, which is about 7 times of magnitudes larger than 1 sun ($\sim 1.7 \times 10^{17}$ photons $\text{cm}^{-2} \text{ s}^{-1}$), <https://doi.org/10.1038/s41586-022-05268-x>, already is strong enough for the perovskite decomposition even just consider the average intensity.

3. It is unreasonable that silicon ions from glass can get through the FTO layer to the perovskite layer. And for the response to Reviewer 1 comment 2, the authors admitted this point:

‘While we do not expect silicon ions to migrate across the FTO and gold layers, the observed enhancement likely suggests an enrichment of silicon in FTO and even gold within the exposed region.’

But in the revised manuscript, on page 13 and figure 4, the authors still claimed:

‘...induce the migration of isovalent tin and silicon ions from the substrate into the sample....’

4. For figure 5, CL analysis of MAPbI₃ film, no significant distribution difference was observed in Component #1 and #2 for different illumination times, as well as for w. or w/o laser illumination. These results are against what they claimed in the first version. If the different conclusion is just related to different laser intensities, it is still evidence that this is a light-induced decomposition process, whether the decomposition product is PbI₂ or not.

5. On page 16, the authors claimed that the recovery of degraded FAPbI₃ sample is a beneficial impact on ion migration. Since the FAPbI₃ cubic (black) phase is unstable at room temperature and can spontaneously transfer to the yellow phase quenching the PL signal, this phenomenon is more likely because the laser-induced heat caused a phase transition from the yellow phase to the black phase.

Revision Report NCOMMS-22-41216 #2

Mapping the Pathways of Photo-induced Ion Migration in Organic-inorganic Hybrid Halide Perovskites

T. Kim et al.

Response to Reviewer 3:

Comments: *Since the authors' responses didn't give more strong evidence to support their conclusion, I insist this is a decomposition process rather than ion migration, so I cannot recommend publishing this manuscript on Nature Communications.*

Response: Although we greatly appreciate their constructive criticism, we respectfully disagree with Reviewer 3's assessment that our observation was due to thermal decomposition, not ion migration. In particular, **we strongly believe that it is unfair to ignore the overwhelming evidence supporting ion migration while only focusing on one criterion (reversibility) that can not be the single defining feature for ion migration.** Our observed ion migration process was irreversible because the vacuum was the sink for iodine ions after they migrated through the sample. After they escaped into vacuum, there is no way to put the iodine ions back into the sample and make the process reversible. This is in contrast to previous electric-field-driven ion-migration processes (e.g. [1]), where excess ions were accumulated at the electrodes and the migration process can be reversed by reversing the electric field direction. **This key experimental difference determines that it is scientifically impossible for us to show the observed photo-induced ion migration is reversible.** However, this irreversibility due to iodine loss to the vacuum does not rule out ion migration inside the sample.

In fact, we have presented extensive evidence in the paper to support that our observed effect was due to ion migration, not decomposition. We list the most definite evidence below:

1. If it was due to thermal decomposition, how could we observe the long-ranged contrast change in the SEM images (e.g. Fig. 2) way outside the illuminated area? If the contrast is due to decomposition, the observed contrast should be limited to the beam area. Our laser beam size was $50\ \mu\text{m}$, while we observed contrast change on the order of hundreds of micrometers outside the illuminated area. This observation can only be explained by ion migration outside the illuminated area induced by iodine deficiency within the illuminated area.
2. If it was due to thermal decomposition, how could we observe an increased *absolute* count of Pb near the surface of the sample and the characteristic two-peak distribution of Pb? Thermal decomposition would lead to deficiency of iodine but would not change the absolute content of Pb. Our observed clear change in absolute Pb count near the surface is a decisive evidence for vertical migration of Pb ions.
3. If it was due to thermal decomposition, why was there no luminescence signature at all of the decomposition product PbI_2 ? In contrast, we characterized an intentionally decomposed FAPbI_3 sample by environmental exposure and observed very

clear emission from PbI_2 (Fig. S21). In fact, it is well established that PbI_2 is the main solid-state product of thermal decomposition of both MAPbI_3 and FAPbI_3 [2, 3, 4]. The lack of any photo-induced PbI_2 signature is a decisive evidence that thermal decomposition in our photo-exposed samples was minimal.

4. In previous studies [3, 4], thermal decomposition of MAPbI_3 and FAPbI_3 typically occurred at temperatures higher than 100 °C. In our experiments, great care was taken to ensure the beam-induced temperature rise (both transient and steady-state temperature rise) was below 30 K.

In addition, as we explained clearly in our first-round revision, ion migration and decomposition are intrinsically intertwined processes that cannot be distinguished in an absolute sense. **Any decomposition process will lead to a nonuniform distribution of ions that will drive ion migration. In the mean time, any ion migration process can also lead to local changes of the material composition that can be termed “decomposition” if the change is significant.** In our case, due to the lack of spectral signature of the decomposition product PbI_2 , we believe the composition change in our samples due to ion migration was small. Accordingly, we prefer to term the effect “ion migration” rather than “decomposition”.

In order to clarify this point, we have now added a separate paragraph in the revised manuscript to discuss why our observation was due to ion migration, not thermal decomposition.

Comments: 1. *The non-reversible changes in Fig. R1 clearly indicate this is a decomposition process. The authors also admitted this on page 11: “Photo-induced decomposition of FAPbI_3 into PbI_2 and, further, into I_2 vapor leads to an iodine deficiency near the surface, which creates an iodine concentration gradient that drives the diffusion of iodine ions towards the surface.”*

Response: As we explained above, the irreversibility of our observation was due to the nature of photo-induced ion migration: after the iodine ions escaped into vacuum, it is impossible to add them back and make the process reversible. However, the irreversibility alone does not rule out the ion migration process in the sample: the ions migrate into a sink (vacuum) and cannot be recovered in our case. We thank the reviewer for pointing out the inaccurate statement in the first version of our paper. Since we did not observe the spectral signature of PbI_2 in the CL measurements, this statement was clearly wrong and we already modified it in the revision.

Comments: 2. *The laser fluence used in this manuscript is $106.6 \mu\text{J}/\text{cm}^2$, which refers to an excitation density of 2.77×10^{17} photons per cm per pulse, average excitation density of 1.385×10^{24} photons $\text{cm}^{-2}\text{s}^{-1}$, which is about 7 orders of magnitude larger than 1 sun (1.7×10^{17} photons $\text{cm}^{-2}\text{s}^{-1}$), already is strong enough for the perovskite decomposition even just consider the average intensity.*

Response: The reviewer’s estimation was inaccurate. Given the optical fluence of $106.6 \mu\text{J}/\text{cm}^2$ and the photon energy of 2.4 eV, the excitation density in our case is 2.77×10^{14} photons per cm. Given the repetition rate of 5 MHz, namely 200 ns between pulses, this corresponds to an average excitation density of 1.38×10^{21} photons $\text{cm}^{-2}\text{s}^{-1}$. Furthermore, an important effect that the reviewer ignored was the photocarrier recombination between pulses. Given the average lifetime of photocarriers on the order of tens of ns

(e.g. Fig. 2c in [5]) in OIHP thin films, the vast majority of photocarriers will have recombined before the next pulse comes in. Therefore, the actual effective excitation level will be much lower than the simple time-averaged value. **This is the key difference between pulsed-laser and continuous-wave-laser experiments, where the photo-damage can be minimized in pulsed-laser experiments even though the nominal excitation levels are the same.**

Comments: 3. *It is unreasonable that silicon ions from glass can get through the FTO layer to the perovskite layer. And for the response to Reviewer 1 Comment2, the authors admitted this point: "While we do not expect silicon ions to migrate across the FTO and gold layers, the observed enhancement likely suggests an enrichment of silicon in FTO and even gold within the exposed region". But in the revised manuscript, on page 13 and figure 4, the authors still claimed: "...induce the migration of isovalent tin and silicon ions from the substrate into the sample..."*

Response: In our response to Reviewer 1 Comment 2, we were discussing the result of a control experiment, where we added a gold layer between perovskite and FTO. The purpose of this experiment was to rule out the possibility that the change of the perovskite film leads to increased substrate exposure, thus a higher silicon signal. This possibility was indeed ruled out since we did not observe an increase of the gold X-ray signal. As we discussed in the revised manuscript, we only presented the migration of silicon as one possible explanation of the observation, and "we do not have direct evidence to support the presence of silicon and tin ions within the perovskite layer after light exposure at this stage, which will be explored in a future work." We agree with the reviewer that our description of Figure 4 can be misleading in this regard, and we have now further modified the manuscript to clarify this point. We added the following sentences: "As a consequence of the lead ion migration, there is a possibility that the lead vacancies left behind deep inside the sample bulk might induce the migration of isovalent tin and silicon ions inside the substrate and towards the sample, leading to an enhanced silicon and tin X-ray count, although we currently do not have direct evidence of their presence inside the perovskite sample." We have also removed Figure 4 in the revised manuscript to avoid the misunderstanding on silicon and tin migration.

Comments: 4. *For Figure 5, CL analysis of MAPbI₃ film, no significant distribution difference was observed in Component 1 and 2 for different illumination times, as well as for w. or w/o laser illumination. These results are against what they claimed in the first version. If the different conclusion is just related to different laser intensities, it is still evidence that this is a light-induced decomposition process, whether the decomposition product is PbI₂ or not.*

Response: As requested by the reviewers in the first round of revision, we replaced Fig.5 (previously measured on a hybrid sample) with the new data taken on pure MAPbI₃. Although the observation was different from the previous dataset, the difference was due to the different samples and, more importantly, was not against any of our claims. The main purpose of this measurement was to check the presence of PbI₂ and verify that our observed effect was indeed ion migration, not decomposition. It is clear from the data that the laser exposure did not induce any increase of PbI₂ signature. In fact, the PbI₂ emission from isolated locations was strongly suppressed by the laser exposure (as shown in column 6). In addition, the normal CL emissions (components 1 and 2) were not affected much by the laser exposure, as noted by the reviewer, providing further evidence that the laser-induced thermal decomposition in our case was minimal.

Comments: 5. On page 16, the authors claimed that the recovery of degraded FAPbI₃ sample is a beneficial impact on ion migration. Since the FAPbI₃ cubic (black) phase is unstable at room temperature and can spontaneously transfer to the yellow phase quenching the PL signal, this phenomenon is more likely because the laser-induced heat caused a phase transition from the yellow phase to the black phase.

Response: We thank the reviewer for pointing out this alternative explanation. We were aware of the structural phase transition in FAPbI₃, but according to literature, the emission from the “yellow” δ -phase FAPbI₃ is broadband over 400 nm to 650 nm [6], which was clearly not what we observed in our experiment. Instead, the narrow emission around 500 nm observed in our measurement was a clear signature of PbI₂. Moreover, after light exposure, the PbI₂ emission from the light-exposed area was completely suppressed (Fig. S21). This observation also cannot be explained by the yellow-to-black phase transition. We added a discussion in the Supplementary Information about this alternative explanation.

References

- [1] Zhengguo Xiao, Yongbo Yuan, Yuchuan Shao, Qi Wang, Qingfeng Dong, Cheng Bi, Pankaj Sharma, Alexei Gruverman, and Jinsong Huang. Giant switchable photovoltaic effect in organometal trihalide perovskite devices. *Nature Materials*, 14(2):193–198, Feb 2015. ISSN 1476-4660. doi: 10.1038/nmat4150. URL <https://doi.org/10.1038/nmat4150>.
- [2] Jérémy Barbé, Michael Newman, Samuele Lilliu, Vikas Kumar, Harrison Ka Hin Lee, Cécile Charbonneau, Cornelia Rodenburg, David Lidzey, and Wing Chung Tsoi. Localized effect of pbi 2 excess in perovskite solar cells probed by high-resolution chemical–optoelectronic mapping. *Journal of Materials Chemistry A*, 6(45):23010–23018, 2018.
- [3] Lin Ma, Deqiang Guo, Mengting Li, Cheng Wang, Zilin Zhou, Xin Zhao, Fangteng Zhang, Zhimin Ao, and Zhaogang Nie. Temperature-dependent thermal decomposition pathway of organic–inorganic halide perovskite materials. *Chemistry of Materials*, 31(20):8515–8522, 2019.
- [4] Azat F Akbulatov, Vyacheslav M Martynenko, Lyubov A Frolova, Nadezhda N Dremova, Ivan Zhidkov, Sergey A Tsarev, Sergey Yu Luchkin, Ernst Z Kurmaev, Sergey M Aldoshin, Keith J Stevenson, et al. Intrinsic thermal decomposition pathways of lead halide perovskites APbX₃. *Solar Energy Materials and Solar Cells*, 213: 110559, 2020.
- [5] Samuel D Stranks, Giles E Eperon, Giulia Grancini, Christopher Menelaou, Marcelo JP Alcocer, Tomas Leijtens, Laura M Herz, Annamaria Petrozza, and Henry J Snaith. Electron-hole diffusion lengths exceeding 1 micrometer in an organometal trihalide perovskite absorber. *Science*, 342(6156):341–344, 2013.
- [6] Fusheng Ma, Jiangwei Li, Wenzhe Li, Na Lin, Liduo Wang, and Juan Qiao. Stable α/δ phase junction of formamidinium lead iodide perovskites for enhanced near-infrared emission. *Chemical Science*, 8(1):800–805, 2017.